# The AaL$^{plus}$ near-peer teaching program in Family Medicine strengthens basic medical skills—A five-year retrospective study

Simon Schwill[1]☯, Jan Hundertmark[1]☯*, Johanna Fahrbach-Veeser[1], Christiane Eicher[1], Pencho Tonchev[2], Sonia Kurczyk[1], Joachim Szecsenyi[1], Svetla Loukanova[1]

**1** Department of General Practice and Health Services Research, University Hospital Heidelberg, Heidelberg, Germany, **2** Department of Surgery, University Hospital Pleven, Pleven, Bulgaria

☯ These authors contributed equally to this work.
* jan.hundertmark@med.uni-heidelberg.de

**Data Availability Statement:** All relevant data are within the paper and its Supporting Information files.

## Abstract

### Background

Basic medical skills such as history taking and physical examination are essential components of clinical work profiles, but nevertheless have been neglected by conventional pre-clinical curricula. The near-peer-teaching program AaL$^{plus}$ [living anatomy plus] teaches basic medical skills, especially history taking, physical examination, and venepuncture, to preclinical students. It is a highly popular compulsory course in the first four semesters (320 students/year, 9h/semester) at Heidelberg University and ends with a formative Objective Structured Clinical Examination (OSCE) during which students receive structured in-depth feedback on their performance. AaL$^{plus}$ is part of the Department of General Practice's longitudinal curriculum for Family Medicine.

### Objectives

This study aims to assess whether the AaL$^{plus}$ program has positive effects on students' clinical skill development and subjective confidence in history taking, physical examination and venepuncture.

### Methods

From 2015 to 2019, we asked all AaL$^{plus}$ participants to rate the program and self-assess their medical skills on 5-point Likert scales (min 1, max 5). In 4-station OSCEs, trained tutors rated the students' performance in all taught skills using standardized checklists.

### Results

From 2015 to 2019 $n = 1534$ questionnaires returned (response rate = 98.6%, 52.7% females). After course completion, students felt able to take a patient's history (mean 3.97, $SD = 0.75$) and perform physical examinations (means range 3.82–4.36, $SD$s range 0.74–0.89) as well as venepuncture (mean 4.12, $SD = 0.88$). A large majority of students claimed they acquired these skills in the AaL$^{plus}$ program. During OSCE, 81.9% passed anamnesis,

**Funding:** The authors received no individual funding. The AaLplus program is part of the regular curriculum for medical students at the Department of General Practice and Health Services Research, University Hospital Heidelberg, Germany. The Ministry of Science, Research and Art Baden-Württemberg financially supports the program as part of the project on "Promoting early clinical competence-oriented teaching according to the Aalplus-program". We acknowledge financial support by the Deutsche Forschungsgemeinschaft within the Open Access Publishing funding program, by the Baden-Württemberg Ministry of Science, Research and the Arts and by Ruprecht-Karls-Universität Heidelberg. The funders had no role in study design, data collection and analysis, decision to publish, or preparation of the manuscript.

**Competing interests:** The authors of this study have read the journal's policy and have the following competing interests: All authors are involved in the AaLplus program. There are no patents, products in development or marketed products associated with this research to declare. This does not alter our adherence to PLOS ONE policies on sharing data and materials.

93.1% passed physical examination, and 95.4% passed venepuncture (of $n$ = 1556). Students mostly rated the feedback they received during the OSCE as "helpful" or "very helpful" (means for different stations 4.69–4.76, $SD$s 0.50–0.70).

## Conclusions

AaL$^{plus}$ is a positive example of a peer teaching program in the preclinical stage of medical studies. It successfully trains junior students in essential medical abilities and increases their confidence in their skills. A high percentage of students pass the formative OSCE and evaluate it positively. Consistently high ratings indicate the program's routine viability. Further studies are needed to analyze if programs like AaL$^{plus}$ could have an impact on the number of graduates choosing career in Family Medicine.

## Background

Medical training needs to convey the broad set of skills and knowledge necessary for future physicians to autonomously and responsibly carry on their profession. Some basic medical skills, such as taking a patient's history and physical examination, are essential components of most clinical work profiles [1, 2]. However, conventional medical curricula have been shown to result in unsatisfactory basic medical skills upon completion of medical school [3–5]. Moreover, medical students and graduates generally do not feel adequately prepared for their work in terms of communicational and clinical-practical skills [6], even though they enjoy initial examinations without diagnostic appliances and seek more "hands-on" contact to patients [7]. Therefore, curricula reforms worldwide aim to strengthen generalist training and improve the integration of basic medical skills [8–13], one notable example being the 2015 German National Competence-Based Catalogue of Learning Objectives in Medicine [14]. This is especially important for doctors specializing in Family Medicine, where communication skills and physical examination skills are highly necessary [15–17].

Effective teaching of basic medical skills requires small teaching groups for individual practice [18], which puts high financial and organizational demands on faculty staff. One feasible and cost-efficient is peer assisted learning (PAL), also known as (near-)peer tutoring or (near-)peer teaching [19–21]. In PAL, more experienced students (tutors) educate the younger and less experienced ones (tutees). Various studies show that tutors are able to convey skills and knowledge as effectively as faculty staff [22–24]. Further recognized benefits of using peers as teachers include cognitive, social, and economic benefits, as well as the acquisition of teaching, organizational, and interpersonal competencies for peer tutors themselves [20, 25–28]. In general, peer-teaching is well established in health professions, especially in nursing education [29, 30]. However, literature supporting formal and long-term integration of near-peer-teaching programs into medical curricula is limited.

To meet the given demand concerning medical students' practical skill acquisition, the PAL program AaL$^{plus}$ (living anatomy plus [Anatomie am Lebenden plus]) was established by an interdisciplinary team at Heidelberg University in 2011. Since then, it has been operated by the Department of General Practice and Health Services Research [31] and undergone continuous evaluation and further development.

The purpose of this study is to a) assess the AaL$^{plus}$ program's effects on students' abilities and subjective confidence in practical medical skills, b) ensure whether students' practical

medical skills level is sufficient to advance to the clinical stage of medical studies, and c) monitor the long-term success of the near-peer teaching program in real-life conditions, i.e., maintained in broad application after the program's pilot phase. We further discuss the program concept as a viable approach to teaching practical medical skills within a longitudinal Family Medicine curriculum. Based on positive impressions and informal feedback we received as university lecturers and medical educators, we hypothesize a positive effect on students' skill and confidence development. Furthermore, we expect sufficiently high pass rates and consistently positive student ratings.

## Methods

### The AaL$^{plus}$ program

In Heidelberg, the 6-year long medical curriculum, HeiCuMed (*Heidelberger Curriculum Medicinale*), is divided into two preclinical and four clinical years and is attended by 320 students per year. AaL$^{plus}$ is a mandatory subject for all students within the preclinical stage. It is embedded in a longitudinal curriculum for family medicine organized by the Department of General Practice and Health Services Research. The program's main goal is to ensure basic medical abilities in history taking, physical examination, and practical skills for all medical students by the end of the second year. Its strong emphasis on clinical practice contrasts other, predominantly theoretical subjects in the first two years of medical school, such as anatomy, physiology, and biochemistry. In total, AaL$^{plus}$ encompasses 30 hours within two years, with five to six sessions of one to two hours duration in two subsequent weeks each semester. The courses take place in the evening and are aligned with contents and schedules of other curricular blocks, including Medical Psychology lectures, Introduction to Careers in Medicine, and two mandatory one-day visitations in general practice. Additionally, the mandatory clerkship in general practice (four weeks) and the general practice course (two weeks) in the later clinical stage of medical studies directly refer back to AaL$^{plus}$ and the competencies gained within it. Moreover, skills in history taking or physical examinations are required for all upcoming clinical rotations and other clerkships, for example in surgery, internal medicine, or neurology. AaL$^{plus}$ integrates theory, hands-on practice, and tutor feedback in a consecutive curriculum for all four preclinical semesters (Fig 1). To allow for active participation and hands-on practical learning, all session take place in small groups (10 to 12 participants) led by teams of two tutors. In history taking, students learn how to conduct a complete anamnesis within different settings. Lessons are supported by "standardized patients", semiprofessional actors who take on patient roles specifically designed for educational purposes. Standardized patients present symptoms and biographical information in a standardized way; they are selected, trained and supervised within the MediKIT program (Medizinisches Kommunikations- und Interaktionstraining [Medical Communication and Interaction Training]) [32, 33] on a regular basis to ensure an authentic performance and high-quality feedback to students. In physical examination lessons, students first receive brief theoretical introductions to then practice examination techniques on each other with the help of standardized checklists (published as the Heidelberg Standards of Examination [*Heidelberger Standarduntersuchungen*]) [34]. Taught techniques encompass examinations of the vertebral column, heart, lungs, liver, and (since 2017) abdomen and thyroid gland, as well as a short neurological examination (since 2018). Practical skills, including hand disinfection, venepuncture, and blood pressure measurement, are taught on dummies and, on a voluntary basis after written informed consent, on one another. Students further practice the problem-based learning approach with different case vignettes every year.

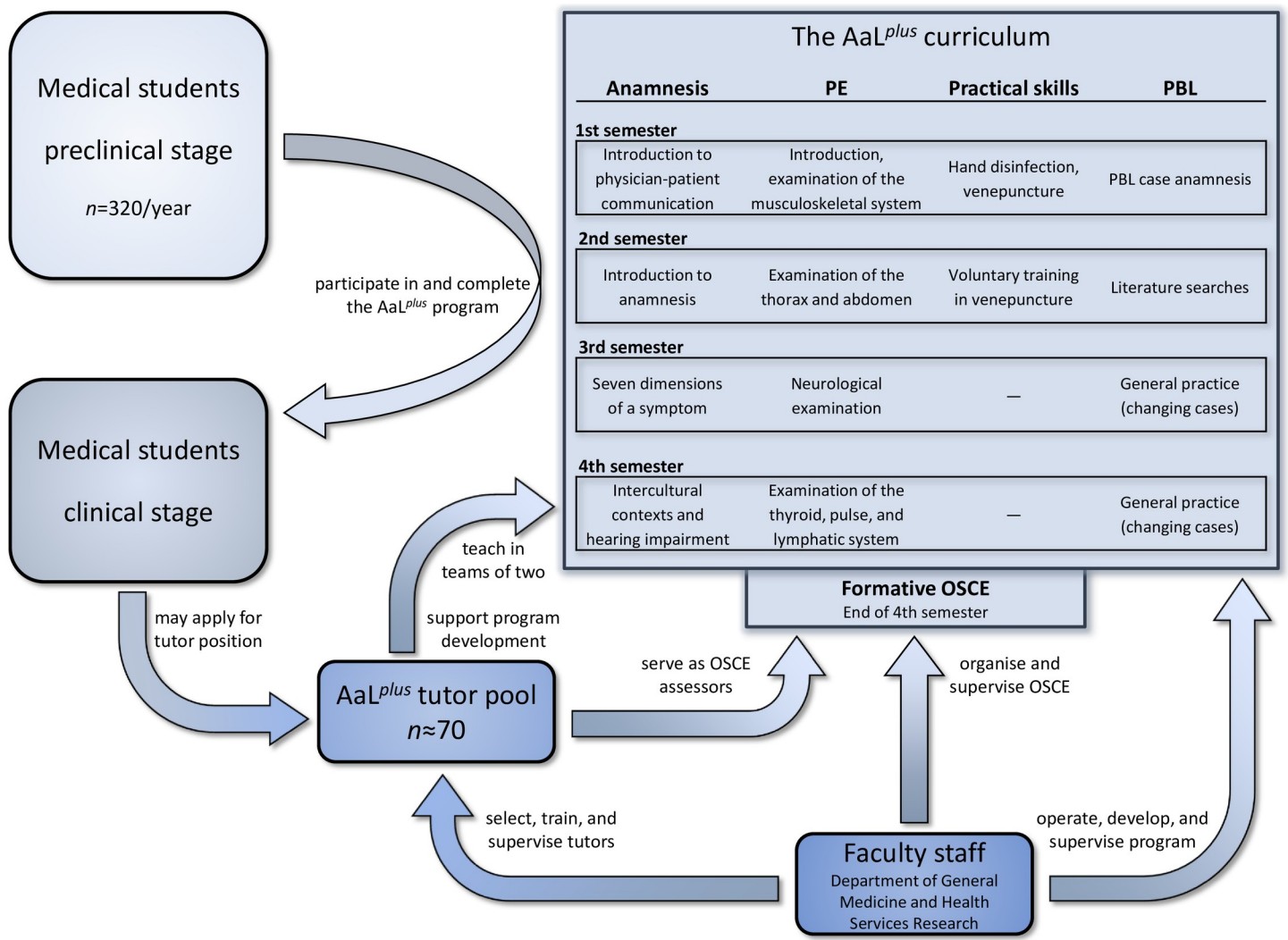

**Fig 1. AaL^plus curriculum and program organization.** Flowchart of student/tutor responsibilities and program organization, including the four-semester curriculum. PBL = Problem-based learning.

In the end of the 4^th preclinical semester, AaL^plus ends with a compulsory but unmarked (formative) OSCE with four OSCE-stations (history taking, two clinical examinations, and venepuncture), which was first established in 2013. Rotation time was 5 min for the examination, 3 min for peer feedback and 1 additional minute for the change to the next station. The assessors used a 25-item standardized checklist for scoring, which was either paper-based (2015) or tablet-computer-based (2016 to 2019).

## Peer tutors

All courses within the AaL^plus program as well as the yearly final OSCE are held by trained peer tutors under supervision by medical staff. The tutors follow a preset thematic structure but autonomously prepare study material, revise and present course content, answer participants' questions, give feedback, facilitate practice phases, and handle group dynamics. They teach their younger peers in teams of two, so that novice tutors are always supported by a senior tutor. Fig 1 also provides an overview on tutor selection and responsibilities. The tutors

are medical students of higher clinical semesters who passed a rigorous selection process based on grades, previous clinical experience, and motivation. In 2018, there were 101 applicants for 21 positions. Before being allowed to teach, new tutors have to complete 40 to 50 hours of preparatory training, in which they acquire general teaching and facilitation skills, team leadership and communication techniques. Special emphasis is put on tutors' abilities to provide structured and high-quality feedback to learners, both in regular AaL$^{plus}$ sessions and in OSCEs. Adding to this, tutors have to pass specific trainings in problem-based learning as well as clinical examinations of heart, lungs, and abdomen every year. These trainings are conducted by educational professionals, general practitioners, and psychologists. These faculty staff members also join in lessons and OSCEs on a regular basis for the purpose of supervising tutors and standardized patients.

In OSCEs, tutors serve as assessors, using standardized checklists for rating and subsequently giving in-depth feedback to examinees [35]. Checklist points include details of history taking (e.g., inquiry into factors aggravating and alleviating pain) or the respective examinations technique (e.g., auscultation locations) and aspects of communicational behavior (e.g., friendliness, empathy). In their feedback to examinees, tutors point out both completed and missing checklist points and furthermore give proposals for improvement. Due to time constraints, standardized patients do not give feedback to examinees. The program has reached a high degree of self-management. A selected team of experienced, highly motivated tutors is involved in the organization and development of course materials, OSCE logistics, and the selection and training of new peer tutors.

## Study design

**Data collection and outcome measures.**   The study was conducted at the Faculty of Medicine at Heidelberg University, Germany. At the end of the academic years of 2015 to 2019, all fourth semester students were invited to participate in a survey. As all data were acquired during a routine, anonymous course-evaluation in medical school, no additional ethics approval was required (Ethics committee of the University of Heidelberg).

Directly after the AaL$^{plus}$ OSCE, all participants completed a 5-minute evaluation sheet, including sociodemographic data and a self-assessment of medical skills. We used a 25-item questionnaire with sociodemographic questions and 5-point Likert scales (1 = not confident at all, 2 = not confident, 3 = balanced, 4 = confident, 5 = absolutely confident). Tutors assessed the participants' OSCE performance using checklists with up to 25 points. 18 points were required to pass, 20 points were counted as a 'fair' result, 22 as 'good', 24 as 'very good'. However, since 2017, assessment for physical examinations was extended to two stations and we used a 30-points anamnesis scale in 2017 and 2018. Therefore, we transformed OSCE ratings to score percentages with 72% required to pass. Some evaluation and skill rating questions were only assessed in later years. Furthermore, to evaluate the consistency of students' ratings and OSCE scores, we calculated intraclass correlations (ICC), defined as the ratio of between-cohort variance to total variance.

## Statistical analysis

All sociodemographic data, self-assessment scores, and OSCE evaluation data were analyzed using SPSS® for Windows (Version 25).

**Table 1. Sociodemographic data.**

| Year | | 2015 | 2016 | 2017 | 2018 | 2019 | 2015–2019 |
|---|---|---|---|---|---|---|---|
| Total number of students | | 301 | 320 | 309 | 301 | 325 | 1556 |
| Number of students participating in the study | | 301 (100%) | 301 (94.1%) | 306 (99.0%) | 301 (100%) | 325 (100%) | 1534 (98.6%) |
| Number of participants providing sociodemographic data | | 298 (97.6%) | 189 (62.8%) | 306 (100%) | 297 (98.7%) | 323 (99.4%) | 1413 (92.1%) |
| Age (years) | mean | 22.2 | 22.0 | 22.2 | 22.1 | 22.3 | 22.2 |
| | SD | 3.27 | 3.28 | 3.23 | 3.90 | 3.75 | 3.49 |
| | range | 18–34 | 18–31 | 18–34 | 18–49 | 18–41 | 18–49 |
| Gender | female | 51.7% | 62.4% | 49.1% | 54.5% | 53.3% | 52.7% |
| | male | 48.3% | 37.6% | 50.9% | 45.5% | 46.7% | 48.3% |
| Previous job training or academic studies | | 29.1% | 29.9% | 29.6% | 27.6% | 31.3% | 29.4% |

Overview on participants' sociodemographic data. Participants are 2nd-year students who completed the AaL^{plus}-curriculum. SD = standard deviation.

## Results

### Participants

Overall, 1534 out of 1556 second-year medical students participating in the OSCE completed the evaluation (response-rate 98.6%, Table 1). Sociodemographic data was available for 92.1% (n = 1413) of participants. Mean age was 22.2 years (SD = 3.49) and 52.7% (n = 745) were female. Overall, 29.4% (n = 415) of students had experienced previous professional training, for instance as medical technical assistants, registered nurses, paramedics or through academic studies. Some medical abilities were only assessed in later years (e.g. pulse measurement or physical examination of the abdomen from 2017 on). In 2016, we asked participants to evaluate via email, which lead to a lower data return. From 2017 on, participant evaluated on site using tablet computers.

### Self-assessment of basic medical skills

Participants assessed their own medical skills on a 5-point Likert scale (Table 2). After participation in AaL^{plus}, the majority of peer-tutees felt confident in taking a patient's history in a structured manner (total mean 3.97, SD = 0.75, n = 1405). Participants felt highly confident that they could perform basic medical skills correctly (total mean hand disinfection 4.57, SD = 0.73, n = 1394; total mean blood pressure measurement 4.61, SD = 0.70, n = 1405; total mean venepuncture 4.12, SD = 0.88, n = 1400; total mean pulse measurement 4.31, SD = 0.79, n = 921). When asked about their physical examination abilities, participants were mostly confident (total mean vertebral column 3.82, SD = 0.84, n = 1405; total mean heart 3.82, SD = 0.83, n = 1391; total mean liver 3.91, SD = 0.82, n = 1403; total mean lungs 3.85, SD = 0.78, n = 1391; total mean thyroid gland 3.82, SD = 0.89, n = 1393; total mean abdomen 4.28, SD = 0.74, n = 911; total mean appendicitis signs 4.36, SD = 0.82, n = 915; total mean lymph nodes 4.03, SD = 0.83, n = 918; total mean neurological examination 4.12, SD = 0.81, n = 619). The high consistency of students' ratings between cohorts is visible in the means shown in Table 2 and further indicated by low ICCs, ranging from < .001 in neurological examination ratings to .063 for vertebral column ratings, mean .025, SD = .023. Table 2 further shows a high consistency between cohorts.

**Table 2. Self-assessment of basic medical skills upon completion of the AaL$^{plus}$-curriculum.**

| I am in the position to perform a… | 2015 | 2016 | 2017 | 2018 | 2019 |
|---|---|---|---|---|---|
| …structured **anamnesis** | 3.78 (0.63) | 3.68 (0.74) | 4.07 (0.70) | 4.14 (0.76) | 4.07 (0.79) |
| | $n = 295$ | $n = 187$ | $n = 305$ | $n = 296$ | $n = 322$ |
| | $p_{Aal} = 93.0\%$ | $p_{Aal} = 88.2\%$ | $p_{Aal} = 88.3\%$ | $p_{Aal} = 92.2\%$ | $p_{Aal} = 89.2\%$ |
| …correct **hand disinfection** | 4.51 (0.68) | 4.62 (0.65) | 4.67 (0.65) | 4.52 (0.79) | 4.56 (0.80) |
| | n = 289 | n = 188 | n = 297 | n = 297 | $n = 323$ |
| | $p_{Aal} = 63.3\%$ | $p_{Aal} = 55.0\%$ | $p_{Aal} = 47.8\%$ | $p_{Aal} = 62.3\%$ | $p_{Aal} = 55.7\%$ |
| …correct **blood pressure** measurement | 4.60 (0.61) | 4.67 (0.61) | 4.65 (0.70) | 4.57 (0.75) | 4.58 (0.77) |
| | n = 295 | n = 189 | n = 295 | n = 296 | n = 319 |
| | $p_{Aal} = 44.9\%$ | $p_{Aal} = 39.0\%$ | $p_{Aal} = 37.2\%$ | $p_{Aal} = 50.2\%$ | $p_{Aal} = 50.2\%$ |
| …**venepuncture** | 3.98 (0.85) | 4.08 (0.83) | 4.27 (0.83) | 4.09 (0.95) | 4.15 (0.89) |
| | n = 290 | n = 188 | n = 302 | n = 297 | n = 323 |
| | $p_{Aal} = 71.0\%$ | $p_{Aal} = 72.5\%$ | $p_{Aal} = 70.5\%$ | $p_{Aal} = 77.9\%$ | $p_{Aal} = 75.6\%$ |
| …correct **pulse** measurement | — | — | 4.29 (0.76) | 4.31 (0.78) | 4.32 (0.83) |
| | | | n = 303 | n = 294 | n = 324 |
| | | | $p_{Aal} = 86.4\%$ | $p_{Aal} = 90.7\%$ | $p_{Aal} = 87.5\%$ |
| …physical examination of the **vertebral column** | 3.66 (0.75) | 3.38 (0.86) | 4.00 (0.78) | 3.95 (0.87) | 3.92 (0.80) |
| | n = 294 | n = 187 | n = 301 | n = 299 | n = 324 |
| | $p_{Aal} = 97.1\%$ | $p_{Aal} = 96.8\%$ | $p_{Aal} = 97.0\%$ | $p_{Aal} = 97.9\%$ | $p_{Aal} = 97.2\%$ |
| …physical examination of the **heart** | 3.70 (0.82) | 3.57 (0.76) | 3.99 (0.79) | 4.05 (0.85) | 4.00 (0.82) |
| | n = 285 | n = 187 | n = 304 | n = 293 | n = 322 |
| | $p_{Aal} = 95.6\%$ | $p_{Aal} = 96.8\%$ | $p_{Aal} = 97.7\%$ | $p_{Aal} = 99.0\%$ | $p_{Aal} = 94.4\%$ |
| …physical examination of the **liver** | 3.70 (0.74) | 3.38 (0.77) | 3.94 (0.76) | 3.99 (0.83) | 3.89 (0.86) |
| | n = 294 | n = 189 | n = 301 | n = 296 | n = 323 |
| | $p_{Aal} = 98.6\%$ | $p_{Aal} = 99.5\%$ | $p_{Aal} = 98.7\%$ | $p_{Aal} = 98.3\%$ | $p_{Aal} = 96.9\%$ |
| …physical examination of the **lungs** | 3.59 (0.72) | 3.65 (0.69) | 3.95 (0.75) | 3.99 (0.81) | 3.98 (0.77) |
| | n = 288 | n = 188 | n = 297 | n = 297 | n = 323 |
| | $p_{Aal} = 95.6\%$ | $p_{Aal} = 95.7\%$ | $p_{Aal} = 95.5\%$ | $p_{Aal} = 98.0\%$ | $p_{Aal} = 95.7\%$ |
| …physical examination of the **thyroid gland** | 3.76 (0.83) | 3.45 (0.92) | 3.86 (0.89) | 4.31 (0.78) | 3.85 (0.92) |
| | n = 283 | n = 187 | n = 297 | n = 294 | n = 322 |
| | $p_{Aal} = 99.3\%$ | $p_{Aal} = 97.8\%$ | $p_{Aal} = 98.7\%$ | $p_{Aal} = 99.0\%$ | $p_{Aal} = 97.8\%$ |
| …physical examination of the **abdomen** | — | — | 4.26 (0.68) | 4.33 (0.76) | 4.26 (0.77) |
| | | | n = 296 | n = 293 | n = 322 |
| | | | $p_{Aal} = 93.6\%$ | $p_{Aal} = 96.9\%$ | $p_{Aal} = 94.7\%$ |
| …check for **appendicitis signs** | — | — | 4.21 (0.86) | 4.41 (0.79) | 4.46 (0.79) |
| | | | n = 295 | n = 298 | n = 322 |
| | | | $p_{Aal} = 89.5\%$ | $p_{Aal} = 92.5\%$ | $p_{Aal} = 91.0\%$ |
| …physical examination of the **lymph nodes** | — | — | 3.99 (0.76) | 4.11 (0.85) | 3.98 (0.86) |
| | | | n = 300 | n = 296 | n = 322 |
| | | | $p_{Aal} = 96.9\%$ | $p_{Aal} = 99.0\%$ | $p_{Aal} = 97.2\%$ |
| …short **neurological** examination | — | — | — | 4.13 (0.80) | 4.11 (0.82 |
| | | | | n = 297 | n = 322 |
| | | | | $p_{Aal} = 96.9\%$ | $p_{Aal} = 95.0\%$ |

Participants' mean self-assessed competence in various medical skills. Ratings on a 5-point Likert scale: 1 = not confident at all, 2 = not confident, 3 = balanced, 4 = confident, 5 = absolutely confident. Standard deviations in parentheses. $n$ = number of answers for respective category. $p_{Aal}$ = percentage of participants who claim to have acquired the respective skill within the AaL$^{plus}$-curriculum.

**Table 3. Overview on participants' OSCE performance.**

| OSCE station | | 2015 | 2016 | 2017 | 2018 | 2019 | All years |
|---|---|---|---|---|---|---|---|
| Number of participants | | 301 | 320 | 309 | 301 | 325 | 1556 |
| Anamnesis | Score *SD* | 78.3% (9.2) 81.7% | 75.9% (9.5) 74.4% | 80.0% (10.1) 81.2% | 81.4% (9.1) 84.7% | 81.4% (10.5) 87.7% | 79.4% (9.9) 81.9% |
| | passed CI | [64.0%, 92.0%] | [60.0%, 92.0%] | [60.0%, 95.0%] | [63.7%, 93.3%] | [60.0%, 96.0%] | [60.0%, 93.3%] |
| Venepuncture | Score *SD* | 86.2% (9.3) 96.0% | 85.4% (10.3) 91.9% | 89.0% (8.5) 95.8% | 89.4% (8.4) 98.0% | 87.9% (9.5) 95.4% | 87.6% (9.3) 95.4% |
| | passed CI | [72.0, 100%] | [64.0%, 100%] | [72.0%, 100%] | [72.4%, 100%] | [72.0%, 100%] | [72.0%, 100%] |
| Physical examination | Score *SD* | 88.1% (10.1) 92.0% | 82.5% (12.1) 85.9% | 86.9% (8.3) 94.8% | 89.8% (6.8) 98.7% | 88.3% (8.5) 95.1% | 87.1% (9.7) 93.3% |
| | passed CI | [68.0%, 100%] | [64.0%, 96.0%] | [70.0%, 98.0%] | [78.0%, 99.8%] | [70.6%, 98.0%] | [68.0%, 100%] |
| Total | Score *SD* | 84.2% (6.8) 74.4% | 76.6% (12.8) 61.2% | 85.4% (6.6) 75.4% | 87.3% (5.9) 81.7% | 86.5% (6.9) 81.5% | 84.0% (9.1) 74.8% |
| | passed CI | [73.3, 93.3%] | [50.7%, 92.0%] | [73.3%, 94.3%] | [77.1%, 96.2%] | [74.0%, 96.0%] | [67.0%, 95.0%] |

Score = mean percentage of achieved points (typically 25). *SD* = score percentage's standard deviation. *n* = number of answers for respective category.

passed = percentage of students who passed the respective OSCE station; a student needs to pass all stations to pass in total. CI = 95% confidence interval of students' OSCE scores.

## Skill acquisition through AaL*plus*

We asked all participants whether they had learned the respective medical skills within the AaL*plus* program or elsewhere (Table 2). On average, 90.3% (of *n* = 1391) reported that they had learned to perform a structured anamnesis through AaL*plus*. More than half of the all participants reported this to be the case for hand disinfection (56.8% of *n* = 1383), less than half for blood pressure measurement (44.8% of *n* = 1371), and 73.7% (of *n* = 1382) of participants for venepuncture. Physical examination skills participants had were almost entirely acquired in the AaL*plus* program (ranging from 91.0% for appendicitis signs to 98.5% for the thyroid gland).

## OSCE results

OSCE ratings were available for 1556 students (Table 3). In the 'anamnesis' OSCE-station, participants achieved a mean score of 79.4% (*SD* = 9.9) and 81.9% of students passed. In the 'venepuncture' station, the mean score was 87.1% (*SD* = 9.7) and 95.4% of students passed. In 'physical examination', students achieved a mean score of 87.1% (*SD* = 9.7) and 93.3% passed. Students' total OSCE score was 84.0% (*SD* = 9.1) and 74.8% passed. To pass the OSCE as a whole, students had to pass all three stations successfully. OSCE performance of different cohorts did not vary substantially, with ICCs of .045 for anamnesis, .028 for venepuncture, and .067 for physical examination, mean ICC = .047.

## OSCE evaluation

Participants used 5-point Likert scales (1 = not agree at all, 2 = disagree, 3 = balanced, 4 = agree, 5 = absolutely agree), similar to their self-assessment, to evaluate selected aspects of the OSCE. In general, participants were highly satisfied with their tutors' performance as OSCE assessors (mean 4.90, *SD* = 0.37, *n* = 619). They rated the feedback they received after the OSCE stations as "very helpful" (anamnesis mean 4.76, *SD* = 0.50, *n* = 1231; venepuncture mean 4.74, *SD* = 0.60, *n* = 1233; physical examination mean 4.69, *SD* = 0.70, *n* = 1204). Altogether, students individual subjective learning progress during the OSCE was high (mean 4.33, *SD* = 0.72, *n* = 616).

## Discussion

In this study we demonstrated that the near-peer teaching-program AaL$^{plus}$ — embedded in a longitudinal curriculum of Family Medicine — supports the basic medical skill acquisition of junior medical students. After our two years curriculum, participants felt confident to take a patient's history in a structured manner, perform important basic medical skills, and complete various relevant basic physical examinations. A large majority states to have learned these skills within the program, with the exception of two skills (hand disinfection and blood pressure measurement), which half of the participants had learned elsewhere. In accordance with their high subjective confidence, about three quarters of all participants successfully passed the (formative) OSCE, even though students faced no negative consequence from failing. Pass rates of individual OSCE stations were even higher. The mean score for anamnesis was slightly lower than for basic skills or physical examination, possibly reflecting the fact that taking a patient's complete history is more difficult to learn. Both self-assessed basic medical skills and OSCE results were consistent across cohorts. Moreover, participants' evaluation of feedback received during the OSCE states that their skills even improved through this feedback.

Taken together, participants' consistently high average confidence in various basic medical abilities, their good pass rates in the OSCE, and their high satisfaction with AaL$^{plus}$ as a whole and with the tutors' performance as assessors are further indicators of students' preparedness for their subsequent clinical study stage as well as the program's success and viability.

Prior research has been shown that early practice of basic medical skills is beneficial for medical students' professional development [8, 10, 36, 37]. In *HeiCuMed*, the two preclinical years' study focus is anatomy, biochemistry, medical psychology, physiology, and basics of other sciences. Therefore, AaL$^{plus}$ (together with two one-day visitations in general practice) offers a welcome opportunity to connect theoretical knowledge (e.g. classical anatomy or physician-patient communication models) and practice (physical examination and history taking), with positive results in line with prior research [38, 39].

Qualitative feedback from our program indicates that the hands-on experience and the insights into clinical routine within AaL$^{plus}$ help to improve participants' motivation the first years of medical school and strengthens their self-confidence. The extents of basic medical skill training in medical schools vary widely between countries. While practical medical experience forms an integral part of medical curricula in Anglo-Saxon or Scandinavian countries [11, 12, 40, 41], this is not the case for other countries [42, 43]. Our research is in line with other studies which report positive effects of early practical medical experience not only in diverse medical abilities, but also in student's interest and motivation to further improve their practical skills throughout their time in medical school [44–46].

These positive effects are likely to be even more pronounced for AaL$^{plus}$ peer tutors. From an educational perspective, tutors are given the tools and experience to become effective teachers [25–27, 35, 47, 48] and benefit in many further ways from their experience. Their close contact and high engagement with the Department of Family Medicine's staff, for instance in trainings, supervisions, or self-management teams, may even strengthen their interest in Family Medicine as a future career choice.

Another problem common during clinical rotations, when medical student start treating patients, is "trial and error" learning on patients [49, 50], potentially leading to hazardous situations. This can be mitigated by ensuring first skill experience in a simulation-based environment [51]. AaL$^{plus}$, as a program strengthening students' basic medical skills, may therefore indirectly contribute to patient-safety and positive physician-patient-relationships.

Taken together, basic medical skills are fundamental for a successful medical career, especially in Family Medicine [16]. A medical student who is well-trained in basic medical skills,

effective communication, and diagnosing illness is highly prepared for all aspects of the career as a family physician [16, 52–54]. In combination with further training in evidence based medicine, a well-trained family physician is able to protect patients from risks related to over-diagnosis and conserve the health system's resources [55]. The AaL*plus* curriculum plants the seed of success in future family physicians, who can rely on their abilities in basic clinical diagnostics.

## Strengths and limitations

The strengths of this study include the large number of participants analyzed over five consecutive years. More than 98% of all students participated, thus providing highly representative data and a valid confirmation of students' abilities as shown in the OSCE, their reasonably high subjective confidence in clinical skills and their high satisfaction with the AaL*plus* program.

Course evaluations and self-assessments are, however, potentially biased and do not necessarily imply skill acquisition [56, 57]. Nevertheless, our results demonstrate high degrees of subjective preparedness and perceived self-efficacy for the clinical stage of medical studies, which is one goal of the program [58]. Furthermore, even though students had no pressure to succeed in the OSCE, pass rates are relatively high (74.8%), indicating that substantial learning indeed took place within the program.

Unfortunately, because AaL*plus* is compulsory for all medical students attending Heidelberg University, it is not possible to form a control group that would allow comparing the program's effect between participants. Due to small curriculum setup changes from year to year, the validity of direct comparisons between cohorts is limited. Instead, future research could match individual OSCE performance with participants' evaluation to verify the validity of their self-assessments.

## Conclusions

AaL*plus*, a near-peer teaching program under the Department of General Practice and Health Services' supervision, strengthens junior medical students' abilities in history taking, physical examination, and practical skills — crucial elements of many clinical work profiles, especially in Family Medicine. Students achieve reasonably high scores in the formative OSCE by the end of their second year. Furthermore, they report high subjective confidence in their abilities and are highly satisfied with feedback received in the OSCE. Five-year data show consistency in all ratings, indicating the program's routine viability. Further studies are needed to explore if early and positive contact with basic medical ability trainings and Family Medicine within a longitudinal curriculum can impact the number of graduates choosing a career in Family Medicine.

## Supporting information

**S1 Data.**
(SAV)

**S2 Data.**
(SAV)

## Acknowledgments

The authors wish to thank all professional trainers as well as all peer-tutors engaged in the AaL*plus*-program for their great work. We highly appreciate the help of Caitlin Walsh for editing the language of the draft. Furthermore, we are very thankful for a supportive interdisciplinary work at medical faculty of Heidelberg. Finally, we highly appreciate the help of Merle Brunnée, Jürgen Krause, and Sonja Rettenmaier coordinating the AaL*plus*-program and the initiative support in 2013–2015 of Nadja Koehl-Hackert, Katja Goetz, and David Pfisterer.

## Previous presentations

Part of data was presented at the annual meeting of the German Society for General Practice and Family Medicine in September 2017 in Düsseldorf, Germany. Conference proceedings were limited to online publications of presentations abstracts. Another part of data (student feedback on assessors' performance in 2016 and 2017) was published in Schwill *et al.*, 2020 [35].

## Author Contributions

**Conceptualization:** Simon Schwill, Svetla Loukanova.

**Data curation:** Simon Schwill, Jan Hundertmark, Christiane Eicher, Sonia Kurczyk.

**Formal analysis:** Jan Hundertmark, Johanna Fahrbach-Veeser, Christiane Eicher, Pencho Tonchev, Sonia Kurczyk, Svetla Loukanova.

**Funding acquisition:** Svetla Loukanova.

**Investigation:** Simon Schwill, Jan Hundertmark.

**Project administration:** Svetla Loukanova.

**Supervision:** Simon Schwill.

**Visualization:** Jan Hundertmark.

**Writing – original draft:** Simon Schwill, Jan Hundertmark, Johanna Fahrbach-Veeser, Svetla Loukanova.

**Writing – review & editing:** Simon Schwill, Jan Hundertmark, Johanna Fahrbach-Veeser, Pencho Tonchev, Sonia Kurczyk, Joachim Szecsenyi, Svetla Loukanova.

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
