## [Decision Letter · Decision Letter 0]

10 Mar 2020

PONE-D-19-29135

The AaLplus near-peer teaching program in Family Medicine strengthens basic medical skills – A cross-sectional analysis

PLOS ONE

Dear Mr. Hundertmark,

Thank you for submitting your manuscript to PLOS ONE. After careful consideration, we feel that it has merit but does not fully meet PLOS ONE’s publication criteria as it currently stands. Therefore, we invite you to submit a revised version of the manuscript that addresses the points raised during the review process.

We would appreciate receiving your revised manuscript by Apr 23 2020 11:59PM. To enhance the reproducibility of your results, we recommend that if applicable you deposit your laboratory protocols in protocols.io, where a protocol can be assigned its own identifier (DOI) such that it can be cited independently in the future. For instructions see: http://journals.plos.org/plosone/s/submission-guidelines#loc-laboratory-protocols

We look forward to receiving your revised manuscript.

Kind regards,

Samy A Azer, M.D., Ph.D., (USyd), M.Ed. (UNSW), M.P.H. (UNSW),

Academic Editor

PLOS ONE

Journal Requirements:

2. If materials, methods, and protocols are well established, authors may cite articles where those protocols are described in detail, but the submission should include sufficient information to be understood independent of these references (https://journals.plos.org/plosone/s/submission-guidelines#loc-materials-and-methods). In order to improve replicability and reproducibility, please provide supporting materials enabling other teachers and researchers to replicate your teaching intervention such as sample worksheets, a detailed lesson plan or curriculum or other such educational materials. If you include supporting materials, they should not be under a copyright more restrictive than CC-BY.

[All authors are involved in the AaLplus program. All authors declare no further competing interests.]

 i) Please confirm that this does not alter your adherence to all PLOS ONE policies on sharing data and materials, by including the following statement: "This does not alter our adherence to  PLOS ONE policies on sharing data and materials.” (as detailed online in our guide for authors http://journals.plos.org/plosone/s/competing-interests).  If there are restrictions on sharing of data and/or materials, please state these. Please note that we cannot proceed with consideration of your article until this information has been declared.

 ii) Please include your updated Competing Interests statement in your cover letter; we will change the online submission form on your behalf.

4. Please include your tables as part of your main manuscript and remove the individual files. Please note that supplementary tables (should remain/ be uploaded) as separate "supporting information" files.

Additional Editor Comments (if provided):

Dear Dr Jan Hundertmark

Thank you for submitting the above-titled manuscript for consideration for publication by PLOS ONE. Your article has been reviewed by two experts and found not suitable for publication.

If you want us to consider it further, please submit an amended copy addressing all points raised by the two reviewers.

You must submit the following:

1. An amended version of the article addressing all points. All changes made to the manuscript are in red or blue colours and underlined (Marked Copy). Also, we need an unmarked copy as well.

2. A response-letter addressing each point raised by each reviewer in a point-by-point format. You may use it in the form of a table showing question/issue raised by the reviewer, the author's response, page number and line numbers.

Please read the guidelines and follow instructions to authors.

Please give attention to grammar and language used.

Please review each reference and ensure that references follow the journal's guidelines.

Submission of an amended version does not guarantee automatic acceptance. The paper will send for peer-review, then a decision will be made.

Thank you

Professor S Azer

EDITOR PLOS ONE.

Reviewers' comments:

Reviewer's Responses to Questions

**Comments to the Author**

1. Is the manuscript technically sound, and do the data support the conclusions?

Reviewer #1: Partly

Reviewer #2: Yes

2. Has the statistical analysis been performed appropriately and rigorously? 

Reviewer #1: No

Reviewer #2: Yes

3. Have the authors made all data underlying the findings in their manuscript fully available?

Reviewer #1: Yes

Reviewer #2: Yes

4. Is the manuscript presented in an intelligible fashion and written in standard English?

Reviewer #1: Yes

Reviewer #2: Yes

5. Review Comments to the Author

Reviewer #1: PONE-D-19-29135

The AaLplus near-peer teaching program in Family Medicine strengthens basic medical skills – A cross-sectional analysis.

This is a good study and is of interest to medical educators. However, there are issues that need to be addressed by the authors.

Title: I think it is “a retrospective study” not cross-sectional, you could state “an analysis of five-year experience.”

Abstract: State what triggered the study. State the study research questions, not just aims.

Abstract - method: Provide more information such as how long did the participants use AaL plus program in each of the four semesters? What was the OSCE about? Did it cover history taking, clinical examination and procedures? How many stations? Not clear how feedback was given? Was feedback given by students (tutees) or also by standardized patients? Time of distribution of questionnaires? Was it before or after the OSCE?

Abstract- results: State number of females (% of females), state the SD, not just the MEAN, state the confidence interval. Were there differences between the results in the first year when you started the program and the results over the next years? How about maintenance issues, did you compare results in 2016, 2017 vs 2019, what was the trend, p values.

The feedback needs to clearly explained, the way it is written shows that there was no standardization or organization. Please give a clear description in the manuscript (not necessary in the abstract). In the abstract, the sentence needs to be rewritten. Words like “very” should not be used.

Abstract- conclusion should be strengthened.

Background- State what triggered the study, State the research questions. Also, a third objective is to show that the program was maintained across these five years. We need to see if there were differences between when the course started compared to next years, and how the course was maintained and were there differences between the years 2016, 2017 and 2019.

Methods- What do you mean by “all students (tutees) participated in the OSCE required…” This sentence needs to be clearly explained, It is not how “near peer-teaching” was organized, prepared, and how students were trained for this job. I see “AaL plus” repeated many times, giving an impression like a computer-aided program? Explain what do you mean by “AaL plus” very early in the manuscript. Describe the program and how it was developed.

Line 19, “In total, AaL plus encompasses 30 hours within two years, with every….” Again, not clear and should be clearly written, how long each session, how many sessions per week, what are the total hours per semester, how many hours in the two preclinical years. How these sessions matched with the blocks/modes in the curriculum?

How were the standardized patients supervised? Did you train them? Details needed.

Methods- The method section is crowded and not well organized. We have several issues AaL plus program, standardized patients, students, tutees (near-peer teaching), clinical exam, history, procedures, and others. You need to add two coloured flow diagrams explaining issues, (see for example the figures in the paper by Azer SA, et al. BMC Med Educ. 2013 May 24;13:71. doi: 10.1186/1472-6920-13-71). This part needs organization as well.

Omit “cross-sectional”; it is a retrospective study. Briefly show how this program was integrated with other programs in the preclinical years, and then with Family Medicine later in the course. What statistical analysis was used?

We need to see if there were differences between when the course started, how the course was maintained and were there differences between the years 2016, 2017 and 2019.

Discussion and conclusions could be strengthened, in light of suggested changes.

References: Some references are too old. Ref 28 covers the same idea here? Explain how this paper adds to what was studied earlier.

Reviewer #2: What is the role of the medical staff supervisors?

In OSCE, the cases are real or dummies? With clinical finding or without? Who prepares and selects these cases for the exam?

Lines 19-20 from page 11, what is the evidence for this conclusion?

Lines 22-23 from page 11, what is the evidence for this conclusion?

Line 2, page 12, is there evidence that these basic skills are especially fundamental for family medicine rather than other clinical specialties? The same comment applies to lines 6-7 page 13.

How can you explain the discrepancy between low scores in self-assessment for hand disinfection and blood pressure measurements on one hand and the relatively high scores in OSCE on the other hand?

6. PLOS authors have the option to publish the peer review history of their article (what does this mean?). If published, this will include your full peer review and any attached files.

Reviewer #1: Yes: Professor Samy Azer

Reviewer #2: No

---

## [Author Response · Author response to Decision Letter 0]

29 Apr 2020

Dear reviewers,

thank you very much for your extensive review which was very helpful for us to improve the manuscript. We now answer to each of your remarks.

Title: I think it is “a retrospective study” not cross-sectional, you could state “an analysis of five-year experience.”

We agree, thank you. We found a wording that indicates the retrospective nature of the study but does not suggest that our experience with the AaLplus program is limited to five years (which it is not)

Abstract: State what triggered the study. State the study research questions, not just aims.

Thank you for your remark. We rephrased the section and now give substantially more detail about research questions plus hypotheses.

Abstract - method: Provide more information such as how long did the participants use AaL plus program in each of the four semesters? What was the OSCE about? Did it cover history taking, clinical examination and procedures? How many stations? Not clear how feedback was given? Was feedback given by students (tutees) or also by standardized patients? Time of distribution of questionnaires? Was it before or after the OSCE?

We included concise information concerning the first four of your questions into the abstract. As to the latter four questions, we instead expanded on them in the methods section. Thank you for your detailed remarks. 

Abstract- results: State number of females (% of females), state the SD, not just the MEAN, state the confidence interval. Were there differences between the results in the first year when you started the program and the results over the next years? How about maintenance issues, did you compare results in 2016, 2017 vs 2019, what was the trend, p values.

We now state the number of females and the SD in the abstract and briefly address the constant evaluation and further development. As to confidence intervals: All CIs for ordinal data (=students‘ self-rating scores) are either [2, 5] or [3, 5]. Due to lack of informational value to the reader, we suggest to omit these statistics. However, we now report confidence intervals for OSCE scores. 

The feedback needs to clearly explained, the way it is written shows that there was no standardization or organization. Please give a clear description in the manuscript (not necessary in the abstract). In the abstract, the sentence needs to be rewritten. Words like “very” should not be used.

We now explain the feedback given to participants in more detail in the respective section below. We use quotation marks to indicate that „very helpful“ refers to the likert scale level.

Abstract- conclusion should be strengthened.

We included a sentence on the program’s routine viability, better reflecting our research hypotheses.

Background- State what triggered the study, State the research questions. Also, a third objective is to show that the program was maintained across these five years.

Thank you for this remark. We now state our research questions in detail.

We need to see if there were differences between when the course started compared to next years, and how the course was maintained and were there differences between the years 2016, 2017 and 2019.

We believe such inferencial comparisons do not aid substantially to answer our research questions. First, we are interested in providing a long-term view on the program’s effects and less on differences between the respective cohorts. Seconds, our program is constantly reevaluated and improved. Significant differences between cohorts‘ ratings or OSCE scores give us little additional insight, as we cannot validly attribute them to either program improvements, tutors‘ practice effect, changes/improvements in rating scales/rating behavior, cohort history effects, etc. Third, with a sample size of n>1500, inferential comparisons are trivial, because even slightest mean differences are statistically significant. Therefore, we report intraclass correlations for students‘ self-ratings. The low ICC statistics indicate that no further group comparison are helpful. In the discussion section, we now also briefly address this.

Methods- What do you mean by “all students (tutees) participated in the OSCE required…” This sentence needs to be clearly explained, It is not how “near peer-teaching” was organized, prepared, and how students were trained for this job. 

We rephrased the sentence.

I see “AaL plus” repeated many times, giving an impression like a computer-aided program? Explain what do you mean by “AaL plus” very early in the manuscript.

Thank you for your remark. We rephrased many sentences, especially in the section „Skill acquisition through AaLplus“. We did not use a computer-aided program.

Describe the program and how it was developed.

Line 19, “In total, AaL plus encompasses 30 hours within two years, with every….” Again, not clear and should be clearly written, how long each session, how many sessions per week, what are the total hours per semester, how many hours in the two preclinical years. How these sessions matched with the blocks/modes in the curriculum?

We now give much more information on these details.

How were the standardized patients supervised? Did you train them? Details needed.

We now added information about supervision and references about the SP program we cooperate with.

Methods- The method section is crowded and not well organized. We have several issues AaL plus program, standardized patients, students, tutees (near-peer teaching), clinical exam, history, procedures, and others. You need to add two coloured flow diagrams explaining issues, (see for example the figures in the paper by Azer SA, et al. BMC Med Educ. 2013 May 24;13:71. doi: 10.1186/1472-6920-13-71). This part needs organization as well.

We reorganized the section, unified the use of terms, and added a synoptic flow diagram to explain the details of AaLplus in a convenient way.

Omit “cross-sectional”; it is a retrospective study. Briefly show how this program was integrated with other programs in the preclinical years, and then with Family Medicine later in the course.

AaLplus is aligned with other curricular blocks, including Medical Psychology lectures, Introduction to Careers in Medicine, and two mandatory one-day visitations in general practice. As proposed, we have added a few sentences on how the program is embedded into the whole curriculum.

What statistical analysis was used?

We need to see if there were differences between when the course started, how the course was maintained and were there differences between the years 2016, 2017 and 2019.

We used SPSS® for Windows, as stated in the „Statistical analysis“ subsection. Again, we suggest to not use inferential analysis.

Discussion and conclusions could be strengthened, in light of suggested changes.

We reworked the relevant details of the discussion and conclusion sections.

References: Some references are too old. Ref 28 covers the same idea here? Explain how this paper adds to what was studied earlier.

Ref 28 is a detailed (yet slightly outdated) program report. In the references, we now provide a translation for its title: [AaLplus — a preclinical course in history taking and physical examination techniques]

Reviewer #2: What is the role of the medical staff supervisors?

We now explain this in more detail: „These faculty staff members also join in lessons and OSCEs on a regular basis for the purpose of supervising tutors and standardized patients.“

In OSCE, the cases are real or dummies? With clinical finding or without? Who prepares and selects these cases for the exam?

Cases are not real, we use standardized patients. Dummies are only used for venepuncture practice. We added much more details to the standardized patients section, which should now answer your questions: „Lessons are supported by “standardized patients”, trained (amateur)semiprofessional actors who take on patient roles specifically designed for educational purposes. Standardized patients present symptoms and biographical information in a standardized way; they are selected, trained and supervised within the MediKIT program (Medizinisches Kommunikations- und Interaktionstraining [Medical Communication and Interaction Training])“

Lines 19-20 from page 11, what is the evidence for this conclusion?

We now give reference to these statements.

Lines 22-23 from page 11, what is the evidence for this conclusion?

This phrase is rather a hypothesis than e conclusion. We slightly changed the wording to make our line of thoughts clearer.

Line 2, page 12, is there evidence that these basic skills are especially fundamental for family medicine rather than other clinical specialties? The same comment applies to lines 6-7 page 13.

We rephrased and now refer more clearly to Franco et al., 2018 

How can you explain the discrepancy between low scores in self-assessment for hand disinfection and blood pressure measurements on one hand and the relatively high scores in OSCE on the other hand?

We apologize, this must be a misunderstanding, the scores in self-assessment for these two skills are amongst the highest. Please let us know if we can rephrase a section to avoid this misunderstanding.

---

## [Decision Letter · Decision Letter 1]

13 May 2020

The AaLplus near-peer teaching program in Family Medicine strengthens basic medical skills – a five-year retrospective study

PONE-D-19-29135R1

Dear Dr. Hundertmark,

We are pleased to inform you that your manuscript has been judged scientifically suitable for publication and will be formally accepted for publication once it complies with all outstanding technical requirements.

With kind regards,

Samy A Azer, M.D., Ph.D., (USyd), M.Ed. (UNSW), M.P.H. (UNSW),

Academic Editor

PLOS ONE

Additional Editor Comments (optional):

Dear Dr Jan Hundertmark

Thank you for submitting an amended version of the above-titled articles. Both reviewers agreed that the manuscript is suitable for publication.

It is my pleasure to inform you that your article has been accepted in PLOS ONE for publication.

The journal office will be in contact with you.

Best regards,

Prof Samy Azer

Editor PLOS ONE

Reviewers' comments:

Reviewer's Responses to Questions

**Comments to the Author**

1. If the authors have adequately addressed your comments raised in a previous round of review and you feel that this manuscript is now acceptable for publication, you may indicate that here to bypass the “Comments to the Author” section, enter your conflict of interest statement in the “Confidential to Editor” section, and submit your "Accept" recommendation.

Reviewer #1: All comments have been addressed

Reviewer #2: All comments have been addressed

2. Is the manuscript technically sound, and do the data support the conclusions?

Reviewer #1: Yes

Reviewer #2: Yes

3. Has the statistical analysis been performed appropriately and rigorously? 

Reviewer #1: Yes

Reviewer #2: Yes

4. Have the authors made all data underlying the findings in their manuscript fully available?

Reviewer #1: Yes

Reviewer #2: Yes

5. Is the manuscript presented in an intelligible fashion and written in standard English?

Reviewer #1: Yes

Reviewer #2: Yes

6. Review Comments to the Author

Reviewer #1: The authors have addressed all points raised by the reviewer. The manuscript is suitable for publication.

Reviewer #2: The authors addressed the comments adequately. I think, significant improvements have been made. Satisfied with the changes.

7. PLOS authors have the option to publish the peer review history of their article (what does this mean?). If published, this will include your full peer review and any attached files.

Reviewer #1: Yes: Professor Samy A Azer, MD, PhD, MPH, MEd, FRSM, FACG

Reviewer #2: Yes: Saad Mohammed AlShareef

---

## [Editor Report · Acceptance letter]

20 May 2020

PONE-D-19-29135R1 

The AaLplus near-peer teaching program in Family Medicine strengthens basic medical skills – a five-year retrospective study 

Dear Dr. Hundertmark:

I am pleased to inform you that your manuscript has been deemed suitable for publication in PLOS ONE. Congratulations! Your manuscript is now with our production department. 

With kind regards,

on behalf of

Professor Samy A Azer 

Academic Editor

PLOS ONE